# Unsupervised Reinforcement Learning with Verifiable Rewards via First Repeat Criterion

## Abstract

Recent advances in Large language models (LLMs) prove that Reinforcement Learning with Verifiable Rewards (RLVR) can enhance the reasoning capabilities of LLMs only with automatically verifiable signals. However, it is still a challenging and labor-consuming task to collect the ground truth answers for reasoning model, especially in the resource-constrained scenario. In this paper, we investigate the potential of unsupervised reinforcement learning with verifiable reward, and propose **uns-GRPO** framework to improve math reasoning of small LLMs. Firstly, we design an **unsupervised reward model** by generating pseudo answer by first repeat criterion. It treats the first repeated answer in a sequence of generated responses as the ground-truth, which is demonstrated with efficiency and reliability in resource-constrained settings. Secondly, we propose an **adaptive KL regularization** to mitigate the noise introduced by pseudo answer. A unique consistency is observed between pseudo answer confidence and accuracy rewards. Adjusting by the accuracy rewards, the adaptive KL regularization enforces conservative optimization when the confidence is low, while encourages diverse exploration when the confidence is high. Experimental results demonstrate that our unsupervised approach achieves stable improvement across diverse models, training datasets, and evaluation tasks.

## 1 Introduction

Recent advances in reasoning-centric large language models (LLMs) have marked a milestone success in the capabilities of artificial intelligence (AI). Landmark models such as OpenAI's o1 (Jaech et al., 2024) and DeepSeek's R1 (Guo et al., 2025) have achieved state-of-the-art performance on complex reasoning tasks spanning mathematics, programming and scientific problem solving. A crucial component of these advancement is the incorporation of post-training strategies into the model development pipeline. Methods such as supervised fine-tuning (SFT) and reinforcement learning (RL) serve to improve reasoning accuracy and customize models to better fit user preferences.

Throughout these innovations, Reinforcement Learning with Verifiable Rewards (RLVR)(Lambert et al., 2024; Guo et al., 2025; Yue et al., 2025) plays as a critical role, as it enables scalable alignment without costly human supervision. RLVR optimizes pretrained models using simple, automatically computed reward signals, e.g., whether a predicted solution matches the correct answer or whether the format of answer matches the requirement. This approach bypasses the need for costly human annotations while enabling the emergence of complex reasoning behaviors. However, it is still a challenging and labor-consuming task to collect the ground truth answers for computing reward signal, specially in the resource-constrained scenario. Creating high-quality, human-verified question–answer pairs is labor-intensive, requiring significant time, cost, and domain expertise. As a result, researchers rise concerns about the long-term viability of relying on human supervision as the primary source of reasoning supervision.

Recent works focus on unsupervised training without the need for additional gold labels (Shafayat et al., 2025) (Zhao et al., 2025a)(Prasad et al., 2024)(Wang et al., 2025). They utilize the model's self-consistency to infer the ground truth answer, and commonly choose majority voting method to assume the most common output as the ground truth. Despite notable advancements, these methods usually rely on extensive response generation and exhibit quadratic computational complexity, which limits the applicability in resource-constrained scenarios.

To address this challenge, we investigate self-improvement methods for large language models in the resource-constrained scenarios, and propose a **uns-GRPO** framework without reliance on ground-truth annotations. An **unsupervised reward model** is design to obtain pseudo answer from a series of outputs. Instead of the exhaustive enumeration in self-consistency approaches, we follow the early-exit principle and design a pseudo-labeling method

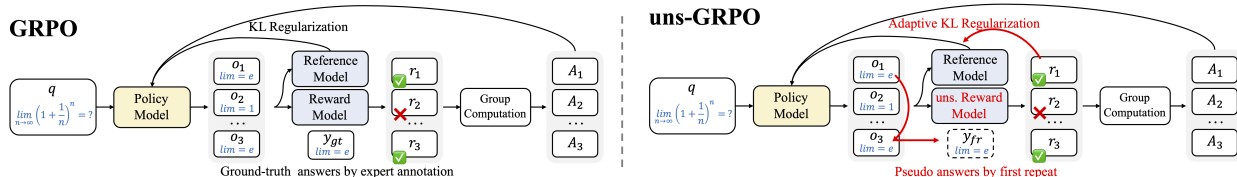

Figure 1: Comparison between the uns-GRPO and the GRPO. Our method generates pseudo answer by first repeat criterion and introduces adaptive KL regularization to mitigate the noise introduced by pseudo labels.

based on the first repeat criterion. It treats the first repeated answer in a sequence of generated responses as the ground truth. Theoretical analysis and experimental results demonstrate that, the first-repeat approach achieves comparable accuracy while enjoying linear computational complexity.

The confidence of the pseudo answer determines the accuracy of reward signal, which further influences the model exploration to downstream task. To mitigate the noise introduced by pseudo answer, we design an **adaptive KL regularization** that dynamically adjusts regularization weight based on label confidence. A phase-transition-like consistency is observed between pseudo answer confidence and the average accuracy reward. Simply put, when the average accuracy reward falls below a certain threshold, the correctness of the pseudo-answers appears random, once it exceeds that threshold, the pseudo-answers are almost always correct. Thus we update the original KL regularization with an adaptive coefficient, to enforce conservative optimization or diverse exploration with an adaptive coefficient $\mathcal{A}$. While the confidence is low, the KL regularization is multiplied with high adaptive coefficient to enforce conservative optimization. In contrast, when confidence is high, the KL term is multiplied with low coefficient to promote exploration.

Building on this design, our uns-GRPO enables a reinforcement learning without ground-truth answers. Experimental results demonstrate that uns-GRPO leads to robust and consistent enhancements in the reasoning ability across diverse models, training datasets, and evaluation tasks.

In summary, our contributions are as follows:

1. We extend the RLVR with an unsupervised paradigm and propose a pseudo-labeling method based on the first-repeat criterion. It treats the first repeated answer in a sequence of generated responses as the ground-truth.

2. We empirically discover the phase-transition-like consistency between pseudo-label confidence and the average accuracy reward. This phenomenon highlights the feasibility of estimating pseudo answer confidence through average accuracy rewards.

3. We propose an adaptive KL regularization framework that dynamically modulates the regularization strength by average accuracy rewards. This mechanism enables the model to maintain stability under noisy supervision by attenuating the influence of low-confidence pseudo answer.

## 2 RELATED WORK

In this section, we first introduce the fundamentals of unsupervised post-training for large model, then provide the theoretical foundations of the first-repeat criterion, finally we give the review of KL regularization in policy optimization.

### 2.1 UNSUPERVISED POST-TRAINING FOR LARGE MODEL

With the rapid development of large models under scaling law(Sharma & Kaplan, 2022)(Hoffmann et al., 2022), the large-scaled and high-quality data become a major bottleneck in terms of cost, time, and expertise(Muennighoff et al., 2025a). To address this challenge, recent works focus on unsupervised post-training without the need for additional gold labels.

A naive approach is to use the model's self-consistency to infer the ground truth answer. Shafayat et. al. leverages the model's self-consistency by majority voting method to infer correctness signal(Shafayat et al., 2025), and they estimating the ground truth answer with the most common solution. Zhao et. al. introduce a self-play approach where

the unified LLM serves as both a proposer and a solver (Zhao et al., 2025a). The system can self-evolves its reasoning ability without gold labels or human-defined queries. Prasad et. al. propose self-consistency preference optimization (SCPO)(Prasad et al., 2024), which is applied at inference time based on multiple sampling in order to find the most consistent answer. Wang et. al. propose a new decoding strategy in chain-of-thought prompting(Wang et al., 2025), it first samples a diverse set of reasoning paths, and then selects the most consistent answer by marginalizing out the sampled reasoning paths.

Besides, some recent research argue that the surrogate rewards can also boost the RLVR in mathematical reasoning without human annotation. Xin et. al. design a length-based rewards to balance promoting comprehensive reasoning and preventing overly long responses(Xin et al., 2025), their method can surpasses the performance of the standard GRPO algorithm with ground truth answers in some certain scenarios. Zhao et. al. replace external rewards in Group Relative Policy Optimization (GRPO) with self-certainty scores(Zhao et al., 2025b). They adopt the self-certainty metric(Kang et al., 2025) and define the scores as the average KL divergence between the uniform distribution and model's each next-token distribution. Shao et. al. point out elicit strong mathematical reasoning in certain models even with spurious rewards that have little correlation (Shao et al., 2025).

Despite notable advancements, these methods usually rely on extensive response generation and introduce quadratic complexity in math verification, which limits the applicability in resource-constrained scenarios. In this paper, we explore unsupervised training with first repeat criterion, and validate the effectiveness and reliability of this method.

## 2.2 First Repeat Criterion

In non-uniform distributions, outcomes with higher likelihood are more prone to early collisions, suggesting that repeated answers may statistically reflect underlying correctness. Building on this idea, several recent works have proposed early stopping criteria based on output repetition to reduce the cost of self-consistency sampling. For instance, Reasoning-Aware Self-Consistency (RASC) (Wan et al., 2025) introduces dynamic stopping strategies that monitor both the agreement among sampled answers and the quality of their underlying reasoning chains. Similarly, Dynamic Voting (Xue et al., 2023) adopts a confidence-driven exit mechanism, when the answer consistency among samples exceeds a preset threshold, the sampling process is terminated early. These approaches demonstrate that repetition-based heuristics can improve both efficiency and accuracy, and offer a foundation for further exploration of more lightweight verification strategies.

In this paper, we extend the unsupervised RLVR with a pseudo answer based on the first repeat criterion. It treats the first repeated answer in a sequence of generated responses as the ground-truth, and is demonstrated with efficiency and reliability in resource-constrained settings.

## 2.3 KL Regularization in Policy Optimization

Kullback–Leibler (KL) divergence is a type of statistical distance which quantifies the dissimilarity between two probability distributions(Van Erven & Harremos, 2014). In policy optimization, KL divergence is a commonly used to limit the divergence between the new policy $\pi_\theta$ and the reference policy $\pi_{\text{ref}}$, offering principled way to constrain stabilize training and facilitate safer exploration (Ouyang et al., 2022).

The specific design of the KL divergence could result in different properties in Policy Optimization. Group Relative Policy Optimization (GRPO)(Shao et al., 2024) adopt $k_3$ estimator (Schulman, 2020) of KL divergence to ensure the divergence constraint is accurately and stably, leading to a robust and effective policy updates. Wang et. al. propose a generalized approach to DPO(Rafailov et al., 2023) by incorporating diverse divergence constraints(Wang et al., 2023). They prove that the efficiency under certain $f$-divergences, including Jensen-Shannon divergence, forward KL divergences and $\alpha$-divergences. To foster more extensive exploration, some recent studies attempt to remove the KL divergence regularization (Yu et al., 2025) (Chu et al., 2025), allowing models to deviate more freely from the reference policy. In summary, a strong KL regularization may suppress model updates, while a weak regularization can encourage the model exploration.

In this work, to mitigate the noise introduced by pseudo-labels, we design an adaptive KL regularization that dynamically adjusts regularization weight based on label confidence. While the confidence is low, the model is regularized more conservatively. When confidence is high, the KL term is relaxed or removed to promote exploration.

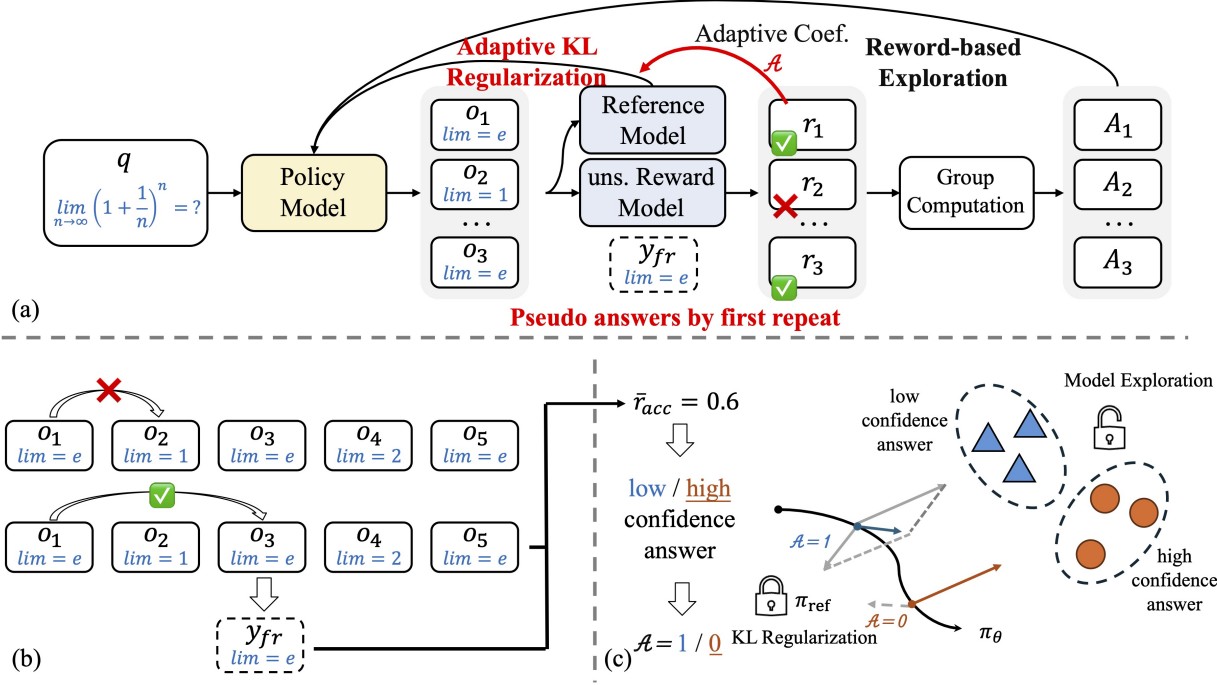

Figure 2: (a) The framework of uns-GRPO. (b) Pseudo answer with first repeat criterion. It treats the first repeated answer in a sequence of responses as the ground truth. (c) Adaptive KL regularization to mitigate the noise introduced by pseudo labels. It enforces conservative optimization or diverse exploration with an adaptive coefficient $\mathcal{A}$.

## 3 METHOD

In this section, we first revisit the optimization objective of GRPO, which can be simplified into two key components, a reward model that promotes exploratory behavior and a KL regularization term that enforces policy stability. Accordingly, we present the optimization objective of our uns-GRPO for the scenario without ground-truth answers (Sec.3.1). Then, we provide the framework and detailed explanation of two core components in uns-GRPO as shown in Fig.2. a) We design an unsupervised verifiable reward by generating pseudo answer with first repeat criterion( Sec.4.3). It treats the first repeated answer in a sequence of generated responses as the ground-truth. which is demonstrated with efficiency and reliability in resource-constrained settings(Appendix A). b) We propose an adaptive KL regularization to mitigate the noise introduced by pseudo labels. A phase-transition-like consistency is observed between pseudo label confidence and accuracy rewards(Sec. 3.3). Thus we control the KL regularization to enforce conservative optimization or diverse exploration with an adaptive coefficient $\mathcal{A}$, which is set as a step function of average accuracy reward $\bar{r}_{acc}$ (Sec.3.4). Finally, we present the overall training procedure of our approach(Sec.3.5).

### 3.1 PRELIMINARIES

Group Relative Policy Optimization (GRPO) (Shao et al., 2024) adapts the PPO (Schulman et al., 2017) framework for training LLMs, notably by eliminating the need for a learned value function. GRPO estimates the advantage $A_i$ at output $o_i$ based on the relative rewards within a group of $G$ outputs $\{o_1, \ldots, o_G\}$ sampled from the old policy $\pi_{\theta_{old}}$ for the same prompt $q$.

$$\mathcal{J}_{\text{GRPO}}(\theta) = \mathbb{E}_{(q,a) \sim \mathcal{P}_q, \{o_i\}_{i=1}^G \sim \pi_{\theta_{old}}(o|q)} \left\{ \frac{1}{G} \sum_{i=1}^{G} \frac{1}{|o_i|} \sum_{t=1}^{|o_i|} \min\left[ r_i A_i, \text{clip}(r_i, 1-\varepsilon, 1+\varepsilon) A_i \right] - \beta \mathbb{D}_{\text{KL}}[\pi_\theta | \pi_{\text{ref}}] \right\},$$
$$(1)$$

where $r_i = \frac{\pi(o_i|q)}{\pi_{old}(o_i|q)}$, and $A_i$ denotes the advantage is computed using a group of rewards.

Accordingly, the optimization objective can be simplified as:

$$\mathcal{J}_{\text{GRPO}} \sim \arg \max_{\theta} \mathbb{E} \left[ \mathbf{R}(q, o, y_{gt}) - \beta \mathbb{D}_{\text{KL}}(\pi_{\theta} \| \pi_{\text{ref}}) \right], \tag{2}$$

where R is the reword function which directly guides the model to make exploration and accomplish goals with human-labeled answer $y_{gt}$. Conversely, the $\mathbb{D}_{\text{KL}}$ constrains the magnitude of policy updates by penalizing the discrepancy between the new policy $\pi_{\theta}$ and reference policy $\pi_{\text{ref}}$, thereby ensuring training stability.

$$\mathcal{J}_{\text{uns-GRPO}} \sim \arg \max_{\theta} \mathbb{E} \left[ \mathbf{R}(q, o, \boxed{y_{fr}}) - \boxed{\mathcal{A}} \beta \mathbb{D}_{\text{KL}}(\pi_{\theta} \| \pi_{\text{ref}}) \right], \tag{3}$$

where $y_{fr}$ denotes the pseudo answer generated by first repeat criterion, and $\mathcal{A}$ denotes the adaptive coefficient of KL regularization depending on the confidence of pseudo answer.

### 3.2 PSEUDO ANSWER BY FIRST REPEAT

To construct pseudo-answers, we begin by generating a set of candidate outputs in response to the input query $q$.

$$\{o_1, o_2, \ldots, o_n\} \sim \pi_{\theta}(\cdot \mid q) \tag{4}$$

Recent efforts in unsupervised reasoning with LLM use signals of self-consistency sampling, which are built on the premise that correct answers are more likely to appear repeatedly across independent reasoning trajectories. Thence the pseudo answer are obtained by a majority voting approch as belowed.

$$y_{mv} \leftarrow \arg \max_{o'} \sum_{i=1}^{n} \mathbf{1}[o_i = o'] \tag{5}$$

However, these methods typically require pairwise comparison among all outputs, resulting in quadratic computational complexity.

In this work, we follow the early stopping principle and obtain pseudo-answers based on the first repeat criterion. As shown in Fig.2(b), it treats the first repeated answer in a sequence of generated responses as ground truth.

$$k = \min\{ i \mid 2 \leq i \leq n, \ \exists j < i: \ o_j = o_i \} \tag{6}$$

$$y_{fr} = o_k \tag{7}$$

By comparing Eq.5 and Eq.7, the first repeat is an efficient method with early-stop strategy, making it applicable in resource-constrained settings. Moreover, we demonstrate its effectiveness and reliability with theoretical proof (Appendix A) and experimental analysis (Secition 4.3 ).

### 3.3 ESTIMATION OF PSEUDO ANSWER CONFIDENCE

As shown in Eq.3, the correctness of the pseudo answer determines the accuracy of reward signal, which in turn influences the model exploration to downstream task. Therefore, estimating the confidence of $y_{fr}$ versus ground-truth answer, becomes a critical issue for effective model optimization.

In this work, we empirically observe a unique consistency between the confidence of pseudo-answers and the average accuracy reward, suggesting the potential of using reward as a proxy for pseudo-answer confidence estimation. We conducted experiments with DeepSeek-R1-Distill-Qwen-1.5B and Qwen2.5-1.5B-Instruct models on the open-rs and GSM8K datasets. Pseudo-answers are obtained via the first repeat method as Eq.7, based on which we calculated the average accuracy rewards $\bar{r}_{acc}$ and evaluated whether the pseudo-answers matched the ground-truth answers. To ensure experimental reliability, each question is repeated test by 100 times.

As Fig.3 shows, the scatter plot clearly illustrates a unique positive correlation between average accuracy rewards and pseudo-answer accuracy, characterized by phase-transition-like behavior, indicating that pseudo answer become

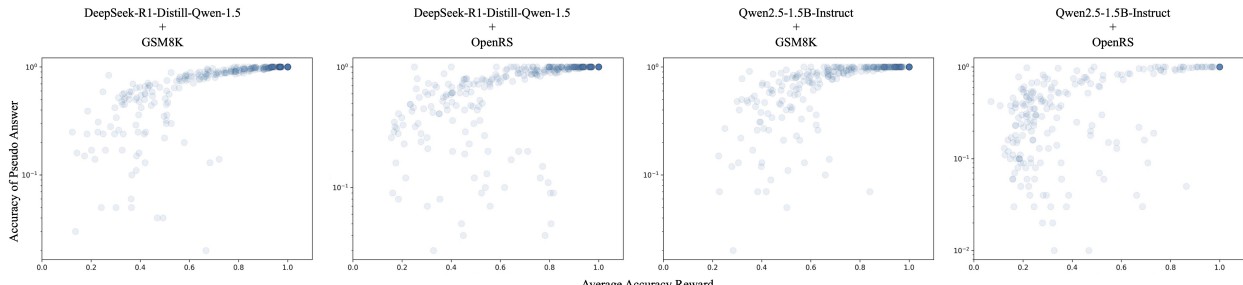

Figure 3: Relation between average accuracy reward and pseudo-answer accuracy. Best Viewed with zoom-in.

consistently accurate once a certain reward threshold is exceeded. At lower accuracy reward levels, pseudo-answer accuracy shows high variability, implying that $\overline{r}_{acc}$ in this range do not reliably predict pseudo-answer correctness. Once the average accuracy reward surpasses the threshold, pseudo-answer accuracy notably increases and becomes stable around 100%. This phenomenon highlights the feasibility of estimating pseudo answer confidence through average accuracy rewards.

Our statistical findings also align with human common intuition. When the $\overline{r}_{acc}$ is high, it indicates that the pseudo-answer obtained by the first-repeat criterion is commonly generated by large model, which typically associated with the ground-truth answer.

### 3.4 ADAPTIVE KL REGULARIZATION

In policy optimization, KL divergence limits the divergence between the new policy $\pi_\theta$ and the reference policy $\pi_{\text{ref}}$, offering principled way to constrain stabilize training and facilitate safer exploration. A strong KL regularization may suppress model updates, while a weak regularization can encourage the model exploration.

Directly using $y_{fr}$ as the pseudo answer may introduce unclean reward signal. To mitigate the noise introduced by pseudo answer, we design an adaptive KL regularization that dynamically adjusts regularization weight. As discussed above, the pseudo answer confidence can be estimated through $\overline{r}_{acc}$. Thus we update the original KL regularization with an adaptive coefficient $\mathcal{A}$, to enforce conservative optimization or diverse exploration with by $\overline{r}_{acc}$ as Eq.8.

$$\mathcal{A} = \begin{cases} 1, & \text{if} \quad \overline{r}_{acc} \in [0, \tau] \\ 0, & \text{if} \quad \overline{r}_{acc} \in (\tau, 1] \end{cases} \tag{8}$$

As Fig.2(c) shows, when the $\overline{r}_{acc}$ is low, the KL regularization is enlarged with high adaptive coefficient to enforce conservative optimization, in condition when $\overline{r}_{acc}$ is high, the KL term is multiplied with low to adaptive coefficient promote exploration.

Furthermore, we present a theoretical analysis to substantiate the robustness of Adaptive KL regularization. The final gradient of our uns-GRPO is formulated as Eq.9,

$$\begin{aligned} \nabla_\theta \mathcal{J}_{\text{uns-GRPO}} &= \nabla_\theta \mathbb{E} \left[ \mathbb{R}(q, o, y_{fr}) - \mathcal{A}\beta \mathbb{D}_{\text{KL}}(\pi_\theta \| \pi_{\text{ref}}) \right] \\ &\sim \mathbb{E} \left[ \nabla_\theta \log \pi_\theta \mathbb{R}(y_{fr}) - \mathcal{A}\beta \nabla_\theta \mathbb{D}_{\text{KL}} \right] \end{aligned} \tag{9}$$

This means the KL divergence regularization $\mathbb{D}_{\text{KL}}$ is effective when $\overline{r}_{acc}$ is low, thus enforces conservative optimization when the pseudo answer is with low confidence.

### 3.5 UNSUPERVISED TRAINING FRAMEWORK

Our uns-GRPO can be easily integrated into various mainstream preference optimization frameworks. During training, we first generate pseudo answer using the first-repeat method. Based on the answers, we compute the average accuracy

reward then adaptively adjust the coefficient of the KL divergence. Finally, we combine the accuracy loss and KL loss to the unified uns-GRPO loss to optimize the model. To enhance reproducibility and implementation clarity, the pseudocode of uns-GRPO is presented in Appendix B.

## 4 EXPERIMENTS

In this section, we present a comprehensive experimental analysis to demonstrate the practical viability of using unsupervised GRPO for mathematical reasoning tasks.

### 4.1 EXPERIMENTAL SETUP

Our uns-GRPO is broadly applicable across a wide range of baseline. We conduct experiments across a variety of models, ranging from 0.5B to 7B parameters. The algorithm is implemented on top of the TRL (von Werra et al., 2020), which is a comprehensive library for post-train foundation models.

During training, the uns-GRPO is designed with following hyperparameters, learning rate of 1e-6, temperature of 0.7 and KL coefficient of 0.0005. Based on empirical observations, we set the threshold parameter $\tau = 0.8$, to encourage exploration when pseudo answer exhibit high confidence. We set 6 outputs per question with a maximum completion length of 32768. The training process is conducted on a cluster of 4 NVIDIA A40 GPUs with 300 steps optimization, except for the 7B model with A100 GPU, making it highly feasible in resource-constrained scenario.

### 4.2 MAIN RESULTS

| Model | AIME24 | MATH-500 | AMC23 | Minerva | OlympiadBench | Average. |
|---|---|---|---|---|---|---|
| *Qwen2.5-0.5B-Instruct* | | | | | | |
| Baseline | 0 | 30.0 | 5 | 6.99 | 7.56 | 9.1 |
| +GRPO | 3.33 | 33.8 | 12.5 | 8.82 | 7.77 | 13.24 |
| +mv-GRPO | 3.33 | 31.4 | 12.5 | 8.46 | 7.41 | 12.62 |
| +uns-GRPO | 3.33 | 32.4 | 12.5 | 8.09 | 7.26 | 12.71(↑ **3.61**) |
| *DeepSeek-R1-Distill-Qwen-1.5B* | | | | | | |
| Baseline | 23.33 | 82.8 | 75.0 | 28.68 | 54.81 | 52.92 |
| +GRPO | 33.33 | 86.6 | 75.0 | 32.35 | 52.44 | 56.54 |
| +mv-GRPO | 33.33 | 85.8 | 77.5 | 30.15 | 52.59 | 55.87 |
| +uns-GRPO | 30.0 | 85.2 | 80.0 | 31.62 | 52.74 | 55.91 (↑ **2.99**) |
| *Qwen2.5-7B-Instruct* | | | | | | |
| Baseline | 13.3 | 75.4 | 52.5 | 33.82 | 37.33 | 42.47 |
| +GRPO | 10.0 | 78.0 | 57.5 | 38.6 | 39.85 | 44.79 |
| +mv-GRPO | 13.33 | 77.6 | 55.0 | 37.13 | 36.89 | 43.99 |
| +uns-GRPO | 10.0 | 77.8 | 55.0 | 37.5 | 37.33 | 43.57 (↑ **1.10**) |
| *DeepSeek-R1-Distill-Qwen-7B* | | | | | | |
| Baseline | 56.67 | 94.0 | 87.5 | 39.34 | 67.11 | 68.92 |
| +GRPO | 56.67 | 94.8 | 95.0 | 41.54 | 67.26 | 71.05 |
| +mv-GRPO | 53.33 | 94.8 | 92.5 | 41.48 | 68.3 | 70.08 |
| +uns-GRPO | 56.67 | 94.6 | 95.0 | 40.44 | 67.85 | 70.91 (↑ **1.99**) |

Table 1: Performance comparison of various models on mathematical benchmarks. The uns-GRPO boosts a stable improvement across diverse models and benchmarks.

Tab.1 compares the performance on a suite of challenging math benchmarks: AIME24, MATH-500, AMC23, Minerva, and OlympiadBench. It highlights the effect of applying uns-GRPO on top of the baseline models. Experimental results show that models with fewer parameters exhibit more significant improvements, likely due to their greater capacity for enhancement. Moreover, for models with same parameter scale, DeepSeek family demonstrates larger gains compared to Qwen family. We hypothesize that this is because DeepSeek models possess stronger inherent reasoning abilities.

### 4.3 PSEUDO ANSWER ANALYSIS

The quality of pseudo answer plays a pivotal role in guiding effective model optimization. In this work, we obtain pseudo-answers based on the first repeat criterion. To compare the accuracy of majority vote method and first repeat method, we conducted experiments with DeepSeek-R1-Distill-Qwen-1.5B and Qwen2.5-1.5B-Instruct models on the GSM8K dataset(Cobbe et al., 2021), open-rs dataset(Dang & Ngo, 2025) and s1K dataset(Muennighoff et al., 2025b).

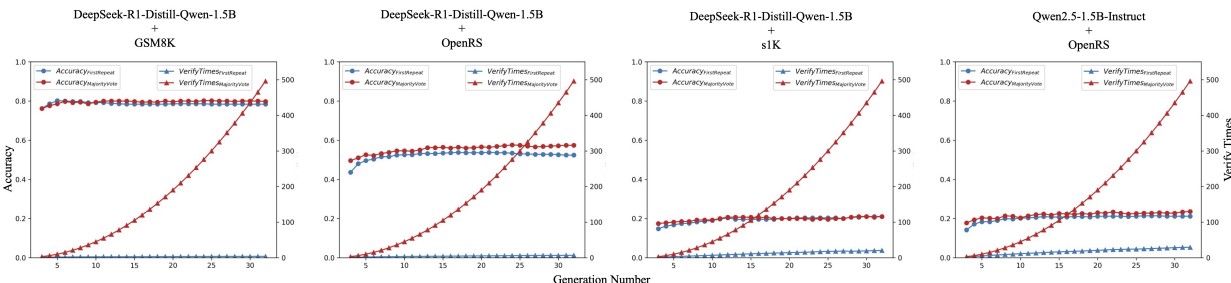

Figure 4: Comparison of pseudo-answer between first repeat and majority vote method. Best Viewed with zoom-in.

As shown in Fig.4, our experiments reveal that the first repeat method achieves comparable accuracy, with only a slight disadvantage. Moreover, as generation sample increases, majority vote perform quadratic complexity due to the need for pairwise comparisons, whereas our first repeat method exhibits only marginal growth in overhead.

As task difficulty increases, from GSM8K to OpenRS and s1K,the accuracy of either the first-repeat and majority vote methods declines. However, their performance remains comparable. While the number of verification attempts required by FR increases with task complexity, it is still significantly lower than that of MV. Moreover, by comparing models with different task difficulty reasoning capabilities, we find that the above conclusion consistently holds. More detailed experiments are attached in Appendix C and Fig. 6 and Fig. 7.

### 4.4 KL REGULARIZATION ANALYSIS

To counteract the adverse impact of erroneous pseudo answer, we propose an adaptive KL regularization that dynamically adjusts regularization strength during optimization.

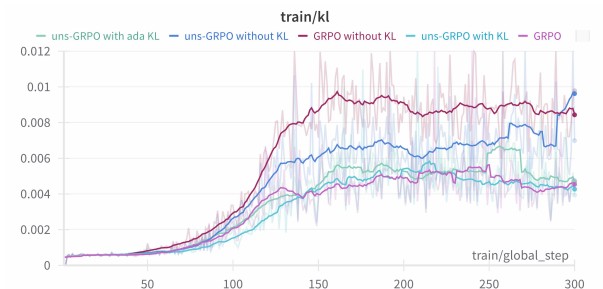

| Method | Accuracy |
|---|---|
| Baseline | 82.8 |
| GRPO w/o KL | 83.2 |
| GRPO w. KL | 86.6 |
| uns-GRPO w/o KL | 81.8 |
| uns-GRPO w. KL | 84.2 |
| uns-GRPO w. reverse Ada. KL | 82.0 |
| uns-GRPO w. Ada. KL | **85.2** |

Figure 5: (a) Left: KL Divergence of different regularization method during training. (b) Right: Accuracy on MATH-500 of different regularization.

As shown in Fig.5, removing KL leads to higher divergence and unstable training, with only marginal even lower performance gains. This phenomenon may be shortcut learning when KL regularization is removed, specially overfitting to the wrong answer.

Directly utilizing KL regularization with uns-GRPO leads to a lower divergence than GRPO, however the improvement is still marginal. We suspect that KL regularization restricts the model from exploration with the correct pseudo answers. Adaptive KL helps uns-GRPO achieve approximate KL divergence to GRPO and best accuracy of 85.2%. Besides, we

conduct a reverse adaptive KL regularization experiment, which restricts the exploration with high confidence answer which encourages exploration with low confidence answer. The performance degradation is significant, indicating that improper KL regularization interferes with the correct learning.

These results highlight the necessity of proper KL control under weak supervision.

## 4.5 ABLATION STUDY

### 4.5.1 TRAINING DATASETS

To further validate the robustness of uns-GRPO, we conduct ablation studies on the training dataset with. The results in Tab.2 indicate that our method delivers consistent performance improvements across diverse data configurations. Notably, the open-rs dataset contributes the greatest performance boost, which we hypothesize is due to the meticulous design and construction of the dataset. Meanwhile, s1K dataset yields limited improvement, likely because its high difficulty prevents the model from producing correct pseudo answer.

| Dataset | baseline | +gsm8k | +deepscaler | +open-rs | +s1K |
|---|---|---|---|---|---|
| Accuracy | 82.8 | 84.8 | 84.6 | **85.2** | 83.4 |

Table 2: Ablation study on training datasets.

### 4.5.2 MODEL TEMPERATURE

As shown in Tab.3, both very low and very high temperatures may degrade the performance. A low temperatures could make the output overly deterministic and rigid, thus result in overfitting problem. In contrast,a high temperatures may introduce too much randomness and lead to difficulty in pseudo answer generation. The optimal performance is achieved at a temperature of 0.7, with the highest accuracy of 85.2%, indicating the best trade-off between determinism and diversity.

| Temperature | 0.1 | 0.3 | 0.5 | 0.7 | 0.9 |
|---|---|---|---|---|---|
| Accuracy | 83.8 | 84.4 | 84.8 | **85.2** | 83.2 |

Table 3: Ablation study on model temperature.

## 5 CONCLUSION

In this work, we aim to enhance the reasoning abilities of reasoning models using unsupervised method. We assume the first repeat answers in a series of responses as the ground truth answer and propose an adaptive KL regularization to mitigate the noise introduced by pseudo answer. Theoretical and empirical analysis validate the effectiveness of uns-GRPO. Furthermore, this study implies that simple pseudo-answer generation methods can be effective in practice. In contrast, enhancing robustness to noisy supervision may serve as a crucial direction for unsupervised reinforcement learning.

ACKNOWLEDGMENTS

**Ethics statement** Our research is a purely methodological investigation of machine learning and does not involve any sensitive individuals. All datasets employed are public available, and this work poses no ethical or privacy concerns.

**Reproducibility statement** To ensure reproducibility, we provide the theoretical proof, pseudocode, and experimental details in Appendix ABC. And the anonymous downloadable source code is at `https://anonymous.4open. science/r/uns-GRPO-65E1`.

**Large Language Model Usage** Authors strictly follow the Policies of ICLR 2026. The research ideas, experiment code are all conduct by authors themselves. LLMs are only used in improving grammar and wording.

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

## A  APPENDIX A: THEORETICAL PROOF OF FIRST REPEAT CRITERION

Given Large Model $\pi_\theta$ and question $q$, we generate $n$ outputs $o_1, ..., o_n \sim \pi_\theta(\cdot|q)$. Let class 1 be the ground truth, and $p_i = \mathbb{P}(o_i \sim \pi_\theta(\cdot|q))$. We define the gap $\Delta := p_1 - \max_{j \geq 2} p_j$, which indicates the probability gap between the correct answer and the most probable incorrect answer. Typically, incorrect answers receive lower predicted probabilities. Therefore, we focus on two representative cases, the correct output and the most probable incorrect outputs, and we have $\Delta > 0$.

## 1. FIRST REPEAT

Aggregate all false labels into a single competitor. First repeat is case of reaching two successes before two failures, which is equivalent to having at least two successes in the first three Bernoulli($p$) trials; a direct counting gives the stated value. we have the uniform lower bound

$$\mathbb{P}\big(y_{fr} = y_{gt}\big) \ \geq \ 3p_1^2 - 2p_1^3 \ . \tag{10}$$

## 2. MAJORITY VOTE

The event that majority vote (with $N$ samples) select class 1 is larger than $\{\mathrm{Bin}(N, p) \geq N/2\}$. Hoeffding's inequality yields

$$\mathbb{P}\big(y_{mv} = y_{gt}\big) \geq \mathbb{P}\big(\mathrm{Bin}(N, p) \geq N/2\big) \ \geq \ 1 - \exp\big(-2N(p - 1/2)^2\big) \ \geq \ 1 - \exp\big(-2N\Delta^2\big) \ . \tag{11}$$

## 3. PROXIMITY BOUND BETWEEN FIRST REPEAT AND MAJORITY VOTE

By the union bound, we have

$$\mathbb{P}\big(y_{fr} = y_{mv} = y_{gt}\big) \ \geq \ \mathbb{P}(y_{fr} = y_{gt}) + \mathbb{P}(y_{mv} = y_{gt}) - 1.$$

Combining equation 10 and equation 11 gives the explicit lower bound

$$\mathbb{P}\big(y_{fr} = y_{mv}\big) \ \geq \ \big(3p_1^2 - 2p_1^3\big) \ - \ \exp\big(-2N\Delta^2\big) \ . \tag{12}$$

**Interpretation.** The term $3p_1^2 - 2p_1^3$ increases monotonically in $p_1$, while the $\exp$ term decays exponentially fast in $N\Delta^2$. Hence, for moderate $N$ and a nontrivial gap $\Delta$, the equal probability is large to 1.

**Conclusion.** Therefore, the first-repeat method and majority voting exhibit comparable accuracy in pseudo-answer generation.

## B APPENDIX B: PSEUDOCODE OF UNS-GRPO

We provide the pseudocode of the uns-GRPO for the help with reproducibility.

---
**Algorithm 1** uns-GRPO
---
**Input:** Training Dataset $\mathcal{X}$, Large Model $\pi_\theta$, Generation Number $n$
**foreach** *Epoch* **do**
    /* Inference Step                                                  */
    Sample minibatch $\mathcal{B} \subseteq \mathcal{X}$
    **foreach** *question $q \in \mathcal{B}$* **do**
        Generate $n$ outputs $o_1, ..., o_n \sim \pi_\theta(\cdot|q)$
        Identify first repeat answer:
          $y_{fr} \leftarrow$ Eq.7
        Compute reward:
          $R(q, o, y_{fr})$
        Compute average accuracy reward:
          $\bar{r}_{acc} = \frac{1}{n} \sum_{i=1}^{\tilde{n}} \mathbf{1}[o_i = y_{fr}]$
        Identify daptive coefficient:
          $\mathcal{A} \leftarrow$ Eq.8
    **end**
    /* Gradient Update step                                         */
    Calculate training loss with Eq.1
    Perform RL gradient update with Eq.9
**end**
---

## C  APPENDIX C: PSEUDO ANSWER ANALYSIS

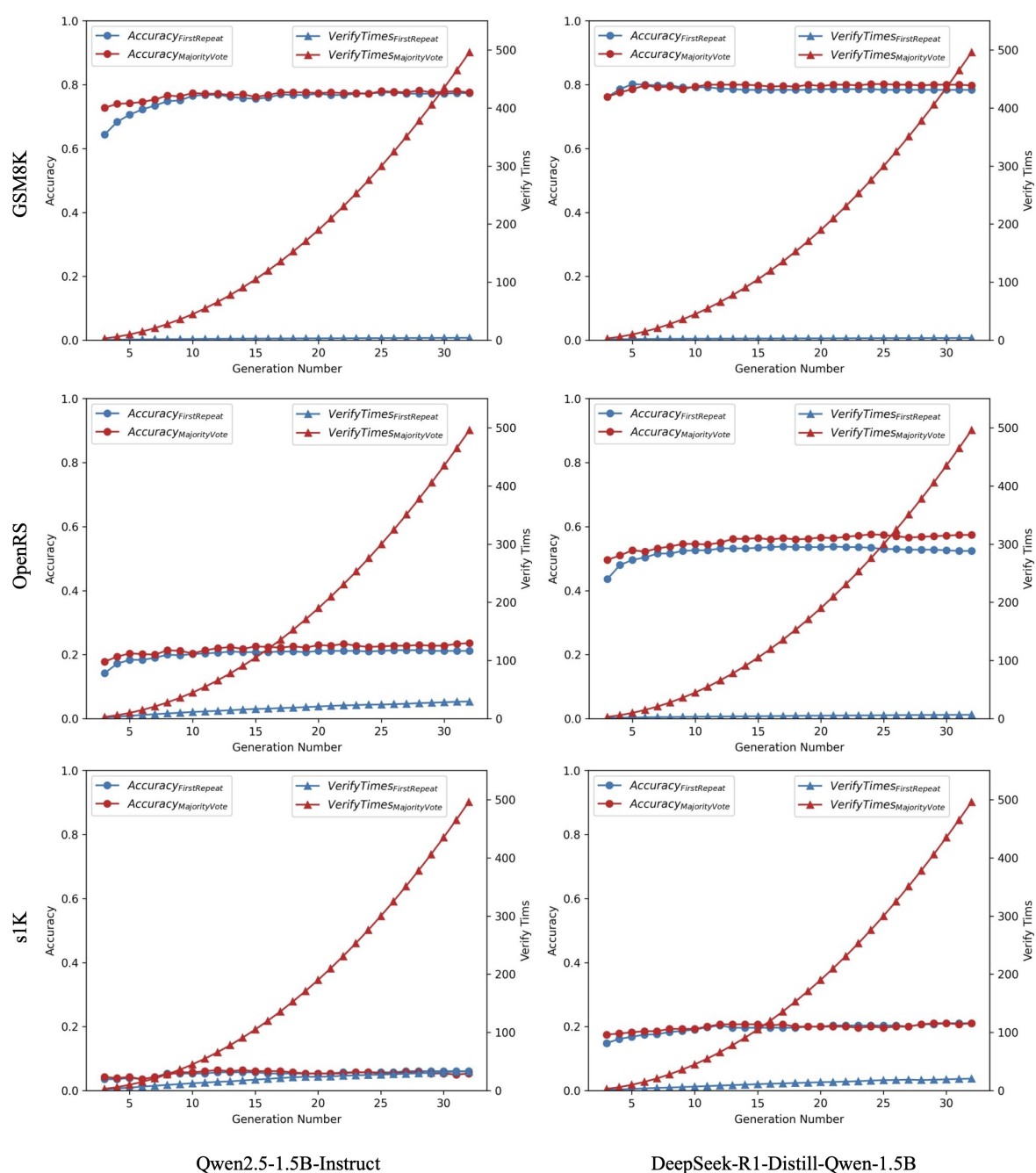

Qwen2.5-1.5B-Instruct       DeepSeek-R1-Distill-Qwen-1.5B

Figure 6: Comparison of pseudo-answer between first repeat and majority vote method. Best Viewed with zoom-in.

We provide more detailed experiments about the pseudo answer in Fig. 6. From left to right, the reasoning ability of the models increases; from top to bottom, the task difficulty increases. As task difficulty increases, from GSM8K to OpenRS and s1K,the accuracy of either the first-repeat and majority vote methods declines. However, their performance remains comparable. While the number of verification attempts required by FR increases with task complexity, it is still

significantly lower than that of MV. Moreover, as reasoning ability increases from Qwen model to DeepSeek model, the accuracy of first repeat method increases and the number of verification decreases.

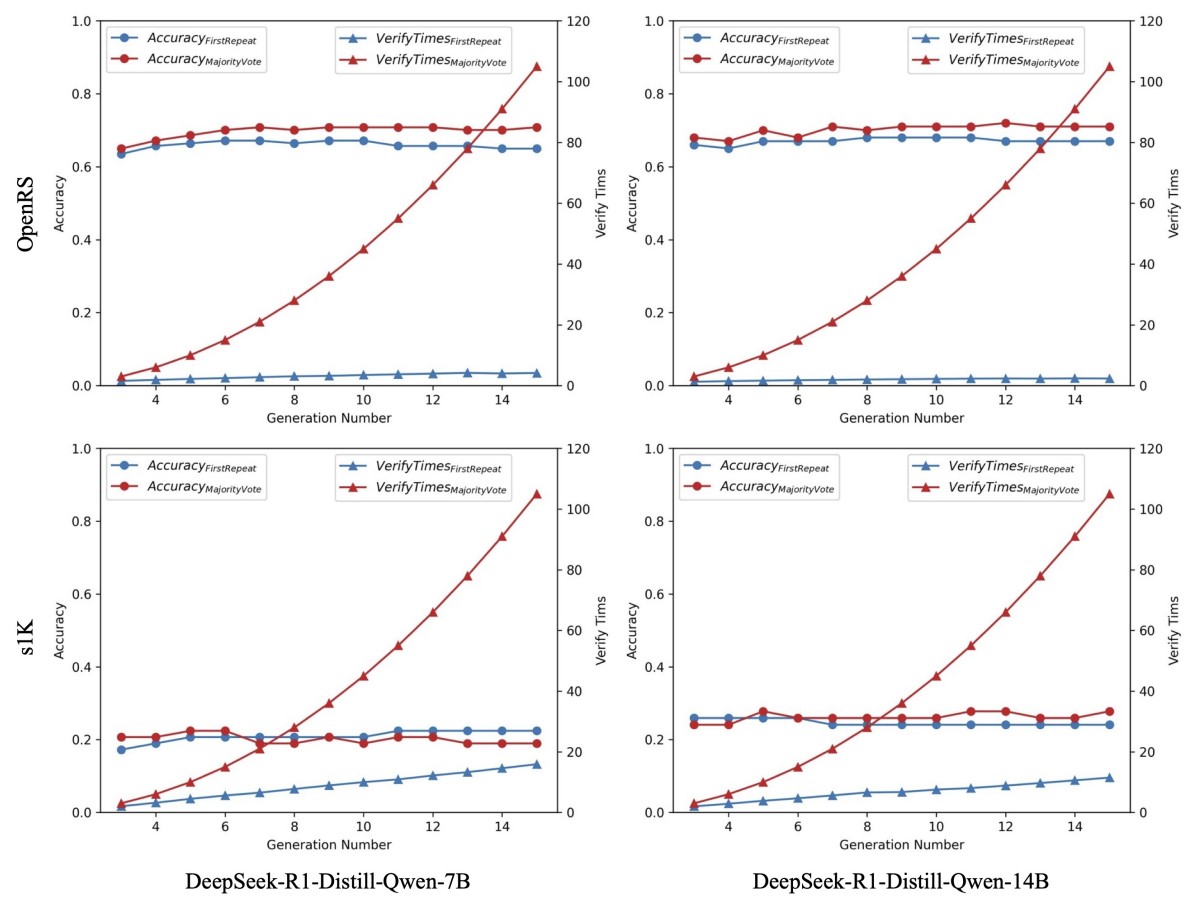

Figure 7: Pseudo answer analysis with 7B and 14B models. Best Viewed with zoom-in.

To verify whether the method can be extended to a large scale , we extended our pseudo answer analysis experiment to include both 7B and 14B models on OpenRS and S1K. As shown in Fig.7, results show that the **performance of first-repeat answer continues to hold at larger scale models**, and the conclusion in Sec. 4.3 remain consistent as we move from small model to large model. This indicates that the first repeat method would still work for larger models.

Experimental results demonstrate that, the first-repeat approach achieves comparable accuracy while enjoying linear computational complexity. This finding further implies that designing elaborate pseudo-labeling may not be such important in unsupervised RL, a simple first repeat method is fair enough.

## D    APPENDIX D: ABLATION STUDY ON $\tau$ IN ADAPTIVE COEFFICIENT

For further sensitivity analysis on the adaptive KL regularization, we supplement study with additional experiments, conducting a finer-grained sampling of $\tau$ as 0.2 and 0.5.

| $\tau$ | 0 | 0.2 | 0.5 | 0.8 | 1 |
|---|---|---|---|---|---|
| Accuracy | 81.8 | 82.2 | 84.0 | **85.2** | 84.2 |

Table 4: Ablation study on $\tau$.

As shown in Tab.4, model accuracy gradually improves wit $\tau$, since uns-GRPO progressively applies KL regularization to low-confidence answers. Performance peaks at $\tau = 0.8$ with an accuracy of 85.2%, then when $\tau$ reaches 1, accuracy drops to 84.2%, indicating that overly relying on the KL constraint restricts the learning freedom from high-confidence answers.

Therefore, both the confidence analysis in Fig.3 and this ablation study consistently support $\tau = 0.8$ as an effective setting. We believe that a more fine-grained or continuous mapping may yield further improvements. However, such designs typically introduce additional computation. Therefore, we intentionally adopt a simple threshold-based design.

# E  APPENDIX E: ABLATION STUDY ON OTHER GRPO VARIANTS

| Model | AIME24 | MATH-500 | AMC23 | Minerva | OlympiadBench | Average. |
|---|---|---|---|---|---|---|
| *DeepSeek-R1-Distill-Qwen-1.5B* | | | | | | |
| Baseline | 23.33 | 82.8 | 75.0 | 28.68 | 54.81 | 52.92 |
| +GRPO | 33.33 | 86.6 | 75.0 | 32.35 | 52.44 | 56.54 (↑ **3.62**) |
| +uns-GRPO | 30.0 | 85.2 | 80.0 | 31.62 | 52.74 | 55.91 (↑ **2.99**) |
| +GPG | 36.67 | 85.8 | 75.0 | 29.78 | 51.58 | 55.77 (↑ **2.85**) |
| +uns-GPG | 30.0 | 85.0 | 77.5 | 29.41 | 52.44 | 54.87 (↑ **1.95**) |
| +DrGRPO | 30.0 | 85.2 | 80.0 | 31.99 | 52.3 | 55.90 (↑ **2.98**) |
| +uns-DrGRPO | 26.67 | 84.2 | 80.0 | 31.25 | 52.15 | 54.86 (↑ **1.94**) |

Table 5: Performance comparison of various models on other GRPO variation. Our first repeat method yeilds a stable improvement with other preference optimization algorithm.

In addition, we apply the first repeat method to other preference optimization algorithms, including DrGRPO and GPG. DrGRPO is designed to achieve an unbiased optimization objective, while GPG directly optimizes the original reinforcement learning objective. Experimental results in Tab. 5 show that our method can be effectively integrated into other preference optimization algorithm, consistently leading to performance improvements. Confirming that the first-repeat pseudo-label provides a high-quality and generalizable supervision signal.

We also observed that GPG and DrGRPO yield slightly smaller improvements over the baseline compared to GRPO. We hypothesize that this may result from their simplified formulations, which make them sensitive to the number of generation samples and the quality of pseudo-answers.

# F  APPENDIX F: WRONG ANSWER BEHAVIOR ANALYSIS

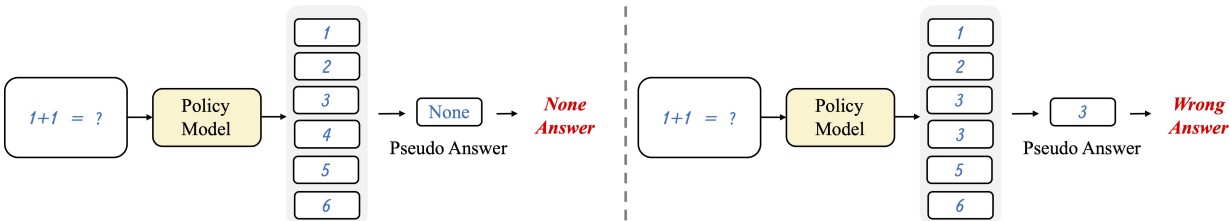

Figure 8: Illustrations of None Answer case and Wrong Answer case. Producing a wrong answer case is a more harmful behavior, which leads to incorrect learning and response.

During pseudo-answer generation, wrong pseudo answer may cause the model to converge to incorrect or misleading knowledge. We conducted additional analysis on the wrong answer behavior across different models. We distinguish

two failure types of pseudo answer: **None Answer** (no repeated value, thus no pseudo-answer) and **Wrong Answers** (an incorrect repeated value).

For example, the question "*What is 1+1?*", a response sequence like [1, 2, 3, 4, 5, 6] contains no repeated value, thus it is a None Answer case. While a response sequence [1, 2, 3, 3, 5, 6] yields a repeated "3", which is a Wrong Answer case. It is evident that producing a wrong answer case is a more harmful behavior.

We introduce a WrongInFailure (WiF) metric to measure the percentage of harmful wrong cases in all failure cases.

$$\text{WiF} = \frac{card(\text{Wrong Answer})}{card(\text{Wrong Answer}) + card(\text{None Answer})} * 100\%. \tag{13}$$

The experiments are conducted with DeepSeek-R1-Distill-Qwen-1.5B/7B/14B model on openrs dataset. And the WiFs are 51.0% / 93.8% / 96.7%. This indicates that **weaker model is much more likely to produce no pseudo answer rather than a wrong pseudo answer**. Interestingly, this observation implies that weaker models may be less likely to converging into wrong answer learning.

