# OpenReview forum: "Unsupervised Reinforcement Learning with Verifiable Rewards via First Repeat Criterion"
_ICLR.cc/2026/Conference — Submitted to ICLR 2026_

### Official Review · Reviewer_H57W · 2025-10-27

**Soundness:** 4
**Presentation:** 3
**Contribution:** 2
**Rating:** 4
**Confidence:** 3

**Summary:**

This paper proposes uns-GRPO, an unsupervised reinforcement learning framework for reasoning models. It replaces ground-truth answers with pseudo-answers generated by a first-repeat criterion, where the first repeated answer among multiple samples is treated as correct. To mitigate noise from potentially wrong pseudo labels, the method adds an adaptive KL regularization: when pseudo-answer confidence is low, KL is strengthened to slow learning; when confidence is high, it is relaxed to allow exploration. Experiments on several math reasoning benchmarks (GSM8K, AIME24, AMC23, Minerva, OlympiadBench) show consistent improvements across models from 0.5B to 7B parameters, with solid ablations on KL settings, datasets, and temperatures.

**Strengths:**

1) Clarity and structure: The paper is clearly written, with straightforward explanations and well-organized experiments.

2) Thorough experiments: The empirical evaluation is extensive—covering multiple model sizes, datasets, KL variants, and temperature settings. Ablations and failure cases (e.g., reverse adaptive KL) are clearly analyzed.

3) Practical significance: The method offers a simple and efficient way to reduce supervision cost in RLVR, achieving stable improvements even in resource-constrained settings.

4) Empirical rigor: Results are consistent across benchmarks, especially showing stronger gains for smaller models.

**Weaknesses:**

1) Limited conceptual novelty:
The core idea is a small modification of prior self-consistency methods. Using the first-repeat heuristic instead of majority vote reduces cost and complexity but does not change the fundamental assumption that the most consistent answer among multiple generations is likely to be correct. The work improves efficiency rather than redefining the underlying learning signal, so the novelty is more engineering-oriented than conceptual.

2) Dependence on model ability:
The framework depends on the base model having a reasonable level of reasoning skill. If the model consistently produces wrong but self-consistent answers, the first-repeat criterion will still select those as pseudo labels. In that case, the model may reinforce its own systematic errors. The adaptive KL term only slows the update when confidence is low; it cannot correct wrong pseudo labels or recover the right reasoning path. Thus, the method improves models that already have moderate capability even though they might not need to be big models, but it may not help very weak ones.

3) Narrow task scope:
All experiments are on math-style reasoning tasks where the answer is explicitly verifiable. The method naturally fits these domains, since correctness can be judged by string equality. However, it is unclear whether the same approach would work in open-ended tasks such as general QA, code generation, or preference learning, where there is no clear notion of a single correct answer. The contribution would be stronger if some results on less verifiable domains were provided.

4) Heuristic KL design:
The adaptive KL is implemented based on a threshold of pseudo-label confidence. This design works in practice but remains a hand-tuned heuristic rather than a principled or continuous uncertainty modeling approach. A smoother or probabilistic control of KL strength might provide more stable or theoretically grounded behavior.

**Questions:**

How sensitive are the results to the specific threshold (τ=0.8) used in adaptive KL? Would a continuous mapping from confidence to KL coefficient perform better?

Could the authors test this framework on non-mathematical reasoning datasets (e.g., code generation or QA) to check generality?

How does the approach behave when the base model is much weaker? does it converge to wrong answers?

Could the pseudo-answer selection benefit from lightweight external calibration to reduce error propagation?

---

> ### Author Response · Authors · 2025-11-21
> **Response to Reviewer H57W (1)**
>
> **W1:Limited conceptual novelty**
>
> AW1: Thanks for the comment. Actually, in a broad sense, pseudo-labeling methods can be almost viewed as different instantiations of the self-consistency principle, including majority vote, cross labeling, and our first-repeat criterion. However, our contribution goes beyond a minor engineering variation in three aspects.
>
> First, the pseudo-answer **generation process is fundamentally different**: first-repeat is an **early-exit strategy**, whereas majority vote is an **exhaustive strategy** that requires generating all samples before making a decision. This early-exit nature makes our approach **applicable to a wider range of reinforcement learning settings** (e.g., Monte-Carlo Tree Search–style rollouts), where stopping early after a reliable repetition can significantly reduce cost.
>
> Second, first-repeat **naturally extends to continuous-output tasks**. For tasks such as object detection or keypoint localization, the output space is continuous, and majority voting becomes ill-defined because exact agreement almost never occurs. In contrast, our framework can be naturally extended by treating outputs within a small numerical tolerance as “repeats”, allowing first-repeat method applicable and efficient in continuous domains.
>
> Third, although first-repeat is a folk wisdom in everyday reasoning, to the best of our knowledge we **make the first attempt to formally analyze and theoretically justify** its rationality in unsupervised learning.
>
> We believe these aspects constitute **a conceptual contribution on unsupervised reinforcement learning**, rather than a purely engineering-oriented modification.
>
> **W2: Dependence on model ability**
>
> AW2: We tested the behavior models with different size and observed an interesting pattern in Table 1, smaller models receive stronger gains. Even a 0.5B model shows a +3.61% improvement, indicating that uns-GRPO does not converge to systematically wrong answers. Instead, the first-repeat heuristic remains effective and provides proportionally larger benefits for weaker models. This indicates that weak models maynot collapse toward wrong answers.
>
> Furthermore, we conducted additional analysis on the wrong answer behavior across different models. We distinguish two failure types of pseudo answer: None Answer (no repeated value, thus no pseudo-answer) and Wrong Answers (an incorrect repeated value).
>
> For example, the question “What is 1+1?”, a response sequence like [1, 2, 3, 4, 5, 6] contains no repeated value, thus it is a None Answer case. While a response sequence [1, 2, 3, 3, 5, 6] yields a repeated “3”, which is a Wrong Answer case.
>
> We introduce a **WrongInFailure** (WiF) metric, WiF = wrong / (wrong + none), to **measure the percentage of harmful wrong cases in all failure cases**. The experiments are conducted with DeepSeek-R1-Distill-Qwen-**1.5B/7B/14B** model on openrs dataset. And the WiFs are **51.0%/93.8%/96.7%**. The WiF metric decreases as the model goes small. This indicates that weak model is much more likely to produce no answer rather than a wrong answer.
>
> The above two findings should support that **weaker smaller models may be less likely to converging into wrong answer learning**.
>
>
>
>
> **W3: Task scope**
>
> AW3: Thank you for raising this important point. We agree that our current formulation relies on exact-match detection, which is naturally suitable for mathematical reasoning where answers follow standardized formats. Actually, **this limitation is not unique to our method, it is shared by prior self-consistency–based approaches**, which rely on exact or equivalence to identify answers.
>
> In addition, we emphasize that the first-repeat criterion offers **a unique advantage over self-consistency methods in continuous-output tasks**. For tasks such as object detection or keypoint localization, the output space is continuous, and majority voting becomes ill-defined because exact agreement almost never occurs. In contrast, our framework can be naturally extended by treating outputs within a small numerical tolerance as “repeats”, allowing first-repeat method applicable and efficient in continuous domains.

---

> ### Author Response · Authors · 2025-11-21
> **Response to Reviewer H57W (2)**
>
> **W4: Heuristic KL design**
>
> AW4: Thank you for the question. The choice of τ = 0.8 is indeed based on empirical observation. As discussed in Sec. 3.3 and Fig. 3, we observe a unique positive correlation with phase-transition-like behavior between average accuracy rewards and pseudo-answer accuracy. Once the average accuracy reward surpasses 0.8, pseudo-answer accuracy notably increases and becomes stable around 100%. Therefore, we set τ = 0.8 to control KL regularization against pseudo-label noise.
>
> In fact, our experiments in **Fig. 5 implicitly include a sensitivity analysis on tau**. The uns-GRPO w/o KL can be regarded as case with tau=0, and uns-GRPO w. KL can be regarded as case with tau=1. Those experiments provide a preliminary ablation study ton tau.
>
> For further rigor, we **supplement study on τ with additional experiments in Appendix D**, conducting a finer-grained sampling of τ as 0.2 and 0.5. The extended experiments reveal a clear pattern: as τ increases, model accuracy gradually improves, since uns-GRPO progressively applies KL regularization to low-confidence answers. Performance peaks at τ = 0.8 with an accuracy of 85.2%, then when τ reaches 1, accuracy drops to 84.2%, indicating that overly relying on the KL constraint restricts the model’s learning freedom from high-confidence answers.
>
> We agree that **a more fine-grained or continuous mapping may yield further improvements**. However, such designs typically **introduce additional computation** and conflicts with the scaling-law motivation. Therefore, we intentionally adopt a simple threshold-based design.
>
> Importantly, **our main contribution is** not the specific form of τ, but **the discovery of a phase-transition phenomenon to confirm the label quality and the demonstration that adaptive KL regulation is beneficial for noisy label learning**. We believe this discovery could provide more meaningful guidance for future research.
>
>
> **Q1: sensitivity analysis on tau**
>
> AQ1: Thanks for your question. We extend the experiment on tau, and both the confidence analysis and the ablation study consistently support tau= 0.8 as an effective setting. We believe a more fine-grained or continuous mapping may yield further improvements, but it goes against with our motivation, and caling-law’s motivation. For detailed discussion, **please refer to AW4**.
>
> **Q2: Test on non-mathematical task**
>
> AQ2: We additionally test our framework on a non-mathematical domain, specifically code generation and question answering, using the **LiveCodeBench** and **GPQA** benchmark via the Lighteval evaluation library[1]. For the DeepSeek-R1-Distill-Qwen-1.5B model, performance on **LiveCodeBench increases from 16.04% to 17.91%** after applying uns-GRPO, meanwhile performance on **GPQA increases from 36.62% to 37.37%**. This confirms that the proposed method is not limited to mathematical reasoning and can also enhance performance in other general tasks.
>
> [1] https://github.com/huggingface/lighteval
>
> **Q3: Will weak models converge to wrong answer**
>
> AQ3:
> We tested the behavior models with different size and observed an interesting pattern in Table 1, smaller models receive stronger gains. Even a 0.5B model shows a +3.61% improvement, indicating that uns-GRPO does not converge to systematically wrong answers. Instead, the first-repeat heuristic remains effective and provides proportionally larger benefits for weaker models. This indicates that weak models maynot collapse toward wrong answers.
>
> Furthermore, we conducted **additional analysis on the wrong answer behavior** across different models. We distinguish two failure types of pseudo answer: None Answer (no repeated value, thus no pseudo-answer) and Wrong Answers (an incorrect repeated value).
>
> For example, the question “What is 1+1?”, a response sequence like [1, 2, 3, 4, 5, 6] contains no repeated value, thus it is a None Answer case. While a response sequence [1, 2, 3, 3, 5, 6] yields a repeated “3”, which is a Wrong Answer case.
>
> We introduce a **WrongInFailure (WiF) metric**,WiF = wrong / (wrong + none), **to measure the percentage of harmful wrong cases in all failure cases**. The experiments are conducted with DeepSeek-R1-Distill-Qwen-**1.5B/7B/14B** model on openrs dataset. And the WiFs are **51.0%/93.8%/96.7%**. The WiF metric decreases as the model goes small.  This indicates that **weak model is much more likely to produce no answer rather than a wrong answer than larger models**.
>
> The above two findings should support that weaker smaller models may be less likely to converging into wrong answer learning.
>
> Actually, the design philosophy of first repeat follows the same spirit as recent scaling law, that is slightly relaxing the accuracy requirements in exchange for dramatically more efficient computation, enabling larger sample budgets and ultimately better overall performance.

---

> ### Author Response · Authors · 2025-11-21
> **Response to Reviewer H57W (3)**
>
> **Q4: Lightweight external calibration for pseudo-answer**
>
> AQ4：
> We agree that lightweight external calibration (e.g., additional scoring signals such as **perplexity score** or **reasoning length**) may further reduce pseudo-label noise and could potentially bring additional gains. This is an interesting direction.  Actually, we have identified a prior study of the relationship between reasoning length and answer correctness[1].
>
> However, our primary design goal is to keep the pseudo-answer selection strictly lightweight, so that **the unsupervised RLVR process remains feasible in resource-constrained settings**. External calibration methods, while helpful, typically introduce additional forward passes or scoring networks, which increase computational cost and inference time.
>
> In the era of large-scale models, we often trade a small amount of precision for improved efficiency in pursuit of scaling laws.
> This trade-off is also the starting point and core motivation of our work.
>
> [1] Between Underthinking and Overthinking: An Empirical Study of Reasoning Length and correctness in LLMs

---

> ### Author Response · Authors · 2025-12-01
> **Extended Experiments on W2: Dependence on model ability**
>
> Dear Reviewer H57W,
>
> We **extended the experiments of WrongInFailure (WiF) metric on s1K** dataset. And the result are listed here.
>
> |  WiF | DeepSeek-R1-Distill-Qwen-1.5B | DeepSeek-R1-Distill-Qwen-7B | DeepSeek-R1-Distill-Qwen-14B  |
> |:---:|:---:|:---:|:---:|
> | openrs | 51.0% | 93.8% | 96.7%  |
> | s1k | 40.1% | 75.6% | 82.0% |
>
> The WiFs for DeepSeek-R1-Distill-Qwen-1.5B/7B/14B model  are 51.0%/93.8%/96.7% on openrs dataset, and 40.1%/75.6%/ 82.0%.  These experimental results are consistent and support each other, **WiF values decline as model capacity decreases or as task difficulty increases**, thus model is much more likely to produce no answer rather than a wrong answer.
>
> The extended experiments further support our conclusion, that is, **weaker models may be less likely to converging into wrong answer learning**.

---

### Official Review · Reviewer_eArp · 2025-10-29

**Soundness:** 3
**Presentation:** 3
**Contribution:** 3
**Rating:** 6
**Confidence:** 4

**Summary:**

In this paper, the authors focus on improving the LLMs' reasoning ability in resource-constrained scenarios. Specifically, they target the situation of being resource-constrained and without ground truth answers, i.e., an unsupervised learning situation. To tackle this problem, the authors designed a framework that applies a pseudo-answer generation mechanism and noise processing methods. Experiments show that the method achieves competitive performance compared to the baselines.

**Strengths:**

1. The motivation of this paper is clear and specific. Rather than generally discussing reasoning ability under resource-constrained scenarios, the authors focus on the specific task of unsupervised learning under resource-constrained scenarios. This focus deepens the discussion depth of this paper.
2. The authors provide sufficient experimental details and hyperparameters for the experiments. In general, the reproducibility of this paper should not be a problem.

**Weaknesses:**

1. In the related work section, the authors mention multiple unsupervised learning methods. However, in the experimental results, like in Table 1, each comparison is only for the baseline and +uns-GRPO. The authors fail to discuss the comparison between their method and other existing methods. This lack makes it hard for the reader to confirm whether the method is the best practice in this field compared to other algorithms.

2. In the experimental setup, the setting of τ = 0.8 seems based on empirical observation. However, the authors fail to provide a sensitivity analysis or ablation study for this important hyperparameter. This raises the question of whether the effectiveness of the method is based on this hyperparameter setting.

**Questions:**

1. Based on Eq. 6, the method relies on finding the exact match answer. This may be feasible for mathematical reasoning because the answers are often in a standardized format. The open question is whether this method can be applied to other problems that require more fuzzy matching. For example, in code generation or logic questions, answers with the same semantics but different expressions will not be recognized as "duplicates".

2. In this paper, the authors focus on models of 7B and below. On a larger scale, like 70b, the self-consistency of LLMs may be high, and the ensemble strategy may not work. Whether the method can be extended to a large scale is also an open question.

---

> ### Author Response · Authors · 2025-11-21
> **Response to Reviewer eArp (1)**
>
> **W1: Comparison with other unsupervised method**
>
> AW1: Thanks for your kind advice. We **add experiments using both GRPO with ground-truth labels (noted as GRPO) and GRPO with majority-vote pseudo-labels (noted as mv-GRPO)** for post-training. As shown in Tab.1, the first-repeat method performs comparably to majority vote, confirming that our lightweight selection strategy is as effective as standard self-consistency–based unsupervised baselines. The GRPO with ground truth rewards performs certainly better, but the performance gap narrows as model size increases. We hypothesize that larger models generate more reliable pseudo-answers, making first-repeat increasingly competitive to ground-truth rewards.
>
> In addition, we **apply the first repeat method to other preference optimization algorithms**, including DrGRPO[1] and GPG[2]. DrGRPO is designed to achieve an unbiased optimization objective, while GPG directly optimizes the original reinforcement learning objective. Experimental results in Tab.5 show that our method can be effectively integrated into other preference optimization algorithm, consistently leading to performance improvements.
>
> Together, these supplementary experiments provide evidence for the robustness and effectiveness of the first repeat method.
>
> [1] Understanding R1-Zero-Like Training: A Critical Perspective
>
> [2] GPG: A Simple and Strong Reinforcement Learning Baseline for Model Reasoning
>
> **W2: ablation study on tau**
>
> AW2: Thank you for the question. The choice of τ = 0.8 is indeed based on empirical observation. As discussed in Sec. 3.3 and Fig. 3, we observe a unique positive correlation with phase-transition-like behavior between average accuracy rewards and pseudo-answer accuracy. Once the average accuracy reward surpasses 0.8, pseudo-answer accuracy notably increases and becomes stable around 100%. Therefore, we set τ = 0.8 to control KL regularization against pseudo-label noise.
>
> In fact, our experiments in Fig. 5 implicitly include a sensitivity analysis on tau. The uns-GRPO w/o KL can be regarded as case with tau=0, and uns-GRPO w. KL can be regarded as case with tau=1. Those experiments provide a preliminary ablation study ton tau.
> For further rigor, we supplement our study with additional experiments in Appendix D, conducting a finer-grained sampling of τ as 0.2 and 0.5. The extended experiments reveal a clear pattern: as τ increases, model accuracy gradually improves, since uns-GRPO progressively applies KL regularization to low-confidence answers. Performance peaks at τ = 0.8 with an accuracy of 85.2%, then when τ reaches 1, accuracy drops to 84.2%, indicating that overly relying on the KL constraint restricts the model’s learning freedom from high-confidence answers.
>
> Therefore, **both the confidence analysis and the ablation study consistently support τ = 0.8** as an effective setting.
>
> **Q1: Feasibility on open question task**
>
> AQ1: Thank you for raising this important point. We agree that our current formulation relies on exact-match detection, which is naturally suitable for mathematical reasoning where answers follow standardized formats. As the reviewer noted, **this limitation is not unique to our method, it is shared by prior self-consistency approaches**, which also rely on exact or normalized equivalence to identify consistent answers.
>
> In addition, we emphasize that the **first-repeat criterion offers a unique advantage over self-consistency methods in continuous-output tasks**. For tasks such as object detection or keypoint localization, the output space is continuous, and majority voting becomes ill-defined because exact agreement almost never occurs. In contrast, our framework can be naturally extended by treating outputs within a small numerical tolerance as “repeats”, allowing first-repeat method applicable and efficient in continuous domains.
>
> **Q2: Performance on large scale model**
>
> AQ2:
> Thank you for raising this question. We agree that examining larger models would further validate the scalability of our approach. Yet, due to computational resource constraints, we were unable to run experiments above 14B. To address your concern, we extended our pseudo answer analysis experiment to include both 7B and 14B models on OpenRS and S1K.
>
> As shown in Fig.7 of the revised manuscript, results show that the performance of first-repeat method continues to hold at larger scale model, and the conclusion in Sec. 4.3 remain consistent as we move from small model to large model. This indicates that the **first repeat method would still work for larger models**.

---

### Official Review · Reviewer_xjoM · 2025-11-01

**Soundness:** 3
**Presentation:** 3
**Contribution:** 3
**Rating:** 6
**Confidence:** 3

**Summary:**

This paper proposes an approach for unsupervised pos-training which samples pseudo answers based on the first-repeat principle, accompanied by a method for estimating the confidence levels of pseudo answers and an adaptive KL regularization to mitigate noise in pseudo labels. The proposed approach is evaluated on a variety of models and datasets and demonstrate consistent advantages.

**Strengths:**

1. The idea of using the first-repeat principle is well motivated, as well as the method for estimating the confidence levels of pseudo labels.
2. The proposed approach is comprehensively tested.
3. The paper is well written and easy to follow.

**Weaknesses:**

1. The texts in figures are very hard to see.
2. The proposed method is not compared with other unsupervised post-training method.

**Questions:**

1. Could you provide empirical evidences for the pros and cons of this method with other unsupervised post training techniques?

---

> ### Author Response · Authors · 2025-11-21
> **Response to Reviewer xjoM (1)**
>
> **W1: texts in figures are small**
>
> AW1:Thank you for your suggestion. To improve readability, we have enlarged the text in the image. Please also note that we have provided larger-sized images in the supplementary materials to ensure reviewers can read them clearly.
>
> **W2: Comparison with other unsupervised method**
>
> AW2: Thanks for your kind advice. We **add experiments using both GRPO with ground-truth labels (noted as GRPO) and GRPO with majority-vote pseudo-labels (noted as mv-GRPO)** for post-training. As shown in Tab.1, the first-repeat method performs comparably to majority vote, confirming that our lightweight selection strategy is as effective as standard self-consistency–based unsupervised baselines. The GRPO with ground truth rewards performs certainly better, but the performance gap narrows as model size increases. We hypothesize that larger models generate more reliable pseudo-answers, making first-repeat increasingly competitive to ground-truth rewards.
>
> In addition, we **apply the first repeat method to other preference optimization algorithms**, including DrGRPO[1] and GPG[2]. DrGRPO is designed to achieve an unbiased optimization objective, while GPG directly optimizes the original reinforcement learning objective. Experimental results in Tab.5 show that our method can be effectively integrated into other preference optimization algorithm, consistently leading to performance improvements.
>
> Together, these supplementary experiments provide evidence for the robustness and effectiveness of the First Repeat method.
>
> [1] Understanding R1-Zero-Like Training: A Critical Perspective
>
> [2] GPG: A Simple and Strong Reinforcement Learning Baseline for Model Reasoning
>
>
> **Q1: Comparison with other unsupervised method**
>
> AQ1: Thanks for the question. We add experiment using GRPO with majority-vote pseudo-labels, and apply the first repeat method to other preference optimization algorithms. These supplementary experiments provide evidence for the effectiveness of the first repeat method. **Please refer to AW2** for details.
>
> The cons of our method may lie in theoretical proof. Although first-repeat is a folk wisdom in everyday reasoning, to the best of our knowledge we make the first attempt to formally analyze and theoretically justify its rationality in unsupervised learning.

---

### Official Review · Reviewer_QgqW · 2025-11-05

**Soundness:** 2
**Presentation:** 2
**Contribution:** 2
**Rating:** 4
**Confidence:** 3

**Summary:**

This paper proposes an unsupervised variant of RLVR, uns-GRPO, for math reasoning problems. Unlike prior unsupervised works which are based on output self-consistency, the proposed approach utilizes a first-repeat criteria: the first answer which is repeated is taken to be the true answer. The benefit of which being computational complexity; rather than requiring a quadratic number of comparisons this approach only requires a linear number with respect to the number of generated outputs. When combined with an adaptive KL regularization scheme, the paper shows improvements over baseline models across several math benchmarks and models.

**Strengths:**

Strengths:
* The answer analysis in Fig. 4 shows some nice computational scaling while yielding comparable accuracy estimates in most cases (although it is not clear whether this trend holds at all points during post-training).
* The proposed method is conceptually simple and should be easy to implement in practice

**Weaknesses:**

Weaknesses:
* Empirical evaluation (Table 1) lacks any baselines other than the baseline model. At the very least this should include the baseline model with ground truth rewards, and would benefit from baselines that use other unsupervised techniques such as majority voting.
* Although the results in Fig. 4 appear impressive on the surface, the dataset seems relatively “easy” since the accuracy plateaus at 5-10 generated samples. This weakens the argument, since the difference in computational time does not significantly differ until higher generation numbers.
* There is an additional hyperparameter that needs tuning: tau (for the adaptive KL). There is no sensitivity analysis over this.
* Minor: Figure text is way too small, and needs to be adjusted.
* Minor: text needs a language/grammar pass.

**Questions:**

Questions:
1. In Fig. 4, at what point during training are the accuracy values computed? Since the model is non-stationary (it changes with every gradient update of GRPO), does this also affect the accuracy values?
2. Does the proposed confidence metric require larger GRPO groups than normal?
3. The language describing the adaptive KL parameter is confusing. Eq. 8 shows it being set to either 0 or 1, but the text describes it as being “enlarged” or “relaxed”, which does not match with the mathematical description. Am I misunderstanding something here?

---

> ### Author Response · Authors · 2025-11-21
> **Response to Reviewer QgqW (1)**
>
> **W1: Experiment with ground truth rewards or other unsupervised rewards**
>
> AW1: Thanks for your kind advice. **We add experiments using both GRPO with ground-truth labels (noted as GRPO) and GRPO with majority-vote pseudo-labels (noted as mv-GRPO)** for post-training. As shown in Tab.1, the first-repeat method performs comparably to majority vote, confirming that our lightweight selection strategy is as effective as standard self-consistency–based unsupervised baselines. The GRPO with ground truth rewards performs certainly better, but the performance gap narrows as model size increases. We hypothesize that larger models generate more reliable pseudo-answers, making first-repeat increasingly competitive to ground-truth rewards.
>
> In addition, we **apply the First Repeat method to other preference optimization algorithms**, including DrGRPO[1] and GPG[2]. DrGRPO is designed to achieve an unbiased optimization objective, while GPG directly optimizes the original reinforcement learning objective. Experimental results in Tab.5 show that our method can be effectively integrated into other preference optimization algorithm, consistently leading to performance improvements.
>
> Together, these supplementary experiments provide evidence for the robustness and effectiveness of the First Repeat method.
>
> [1] Understanding R1-Zero-Like Training: A Critical Perspective
>
> [2] GPG: A Simple and Strong Reinforcement Learning Baseline for Model Reasoning
>
> **W2: First repeat performance on difficult dataset**
>
> AW2:
> Thank you for the insightful comment. As reported in Fig.6, we conduct experiments on datasets with different difficulities, including GSM8K, OpenRS and s1K.The results show that the reliability of the first-repeat criterion still holds, even when the task difficulty is significantly high as s1K.
>
> Importantly, we also observe that the gap between first-repeat and self-consistency is smallest in the 10 to 15 generation range. In such case, majority vote requires 5 to 10 times more verification operations than first-repeat, resulting in substantially higher computational cost. Our choice of 6 generation is due to computational resource limitations.
>
> **W3: Sensitivity analysis on tau.**
>
> AW3: Thank you for the question. The choice of τ = 0.8 is indeed based on empirical observation. As discussed in Sec. 3.3 and Fig. 3, we observe a unique positive correlation with phase-transition-like behavior between average accuracy rewards and pseudo-answer accuracy. Once the average accuracy reward surpasses 0.8, pseudo-answer accuracy notably increases and becomes stable around 100%. Therefore, we set τ = 0.8 to control KL regularization against pseudo-label noise.
>
> In fact, our experiments in Fig. 5 implicitly include a sensitivity analysis on tau. The uns-GRPO w/o KL can be regarded as case with tau=0, and uns-GRPO w. KL can be regarded as case with tau=1. Those experiments provide a preliminary ablation study ton tau.
>
> For further rigor, we supplement our study with additional experiments in Appendix D, conducting a finer-grained sampling of τ as 0.2 and 0.5. The extended experiments reveal a clear pattern: as τ increases, model accuracy gradually improves, since uns-GRPO progressively applies KL regularization to low-confidence answers. Performance peaks at τ = 0.8 with an accuracy of 85.2%, then when τ reaches 1, accuracy drops to 84.2%, indicating that overly relying on the KL constraint restricts the model’s learning freedom from high-confidence answers.
>
> Therefore, **both the confidence analysis and the ablation study consistently support τ = 0.8 as an effective setting**.
>
> **W4: figure text**
>
> AW4: Thank you for your suggestion. To improve readability, we have update the manuscript and enlarged the text in the image. Please also note that we have provided larger-sized images in the supplementary materials to ensure reviewers can read them clearly.
>
> **W5: language/grammar pass**
>
> AW5: Thank you for the suggestion. We carefully revise the manuscript for grammar, clarity, and overall readability.

---

> ### Author Response · Authors · 2025-11-21
> **Response to Reviewer QgqW (2)**
>
> **Q1: Accuracy values computing method**
>
> AQ1：
> We conduct the evaluation in Fig.4 with **offline setting**, using official models available on HuggingFace.
>
> For each question, we randomly generate a fixed number of answers, and then compute the accuracy and number of verification steps for both the First Repeat and Majority Vote strategies. The final results are reported as the average values.
>
> An online evaluation, in which the model is updated with every gradient step, may introduce minor fluctuations in accuracy compared to offline models. Yet we argue that such effects are negligible in our setting. As illustrated in Table 1, even after 300 steps of optimization, the performance does not show an order-of-magnitude improvement relative to the original model, thus the performance remains within a comparable range.
>
> Moreover, online evaluation at every training step is with huge cost. For example, a 1.5B model requires approximately 3 GB of storage, and each evaluation takes about 30 minutes. Given that our method involves training over 300 steps, an online evaluation would demand nearly 1 TB of storage and approximately one week of testing time.
>
> Therefore, we adopt the offline evaluation strategy as a more practical and resource-efficient alternative.
>
> **Q2: GRPO group size for confidence metric**
>
> AQ2:
> We do not need a larger GRPO group size for confidence metric, and the group size is set as 6 in our research. In the original art of GRPO, the group size is set as 64[1]. Meanwhile, some improved variants of GRPO adopt smaller group sizes, such as 16[2] or even 6[3].
>
> Our experimental analysis in Fig.3 shows that a phase transition in confidence typically occurs when the average accuracy exceeds 0.8. As a result, setting the group size greater than 5 is sufficient to capture this signal. Considering computational resource constraints, we finally set the group size to 6.
>
> [1] DeepSeekMath: Pushing the Limits of Mathematical Reasoning in Open Language Models
>
> [2] DAPO: An Open-Source LLM Reinforcement Learning System at Scale
>
> [3] Reinforcement Learning for Reasoning in Small LLMs: What Works and What Doesn't
>
> **Q3: language description of adaptive KL parameter**
>
> AQ3:Thank you for your careful reading. Our original intention was to apply stronger KL constraints in low-confidence cases, as opposed to merely reducing regularization in high-confidence scenarios. And this design aims to suppress noise from incorrect pseudo answers by imposing tighter constraints. To avoid misunderstanding, we have revised the word enlarged/relaxed to multiplied.

---

### Author Response · Authors · 2025-11-16
**Thank You Note**

Hi all,

Thank you for your contribution. We appreciate the reviewer’s careful evaluation and insightful comments.

The authors are now working on supplementary experiments with full dedication, to further strengthen the paper and resolve the raised concerns.

Due to limitations in computational resources, some experiments may not be completed quickly. We sincerely ask for the reviewers’ understanding and patience as we work diligently to complete them.

---

### Author Response · Authors · 2025-11-21
**Response to All Reviewers (1)**

We thank all reviewers for their constructive and insightful feedback. We are very grateful for the recognition shown by reviewers, including:

1 **Motivation is clear**, e.g.,
“The idea of using the first-repeat principle is well motivated” (Reviewer xjoM),
“The motivation of this paper is clear and specific” (Reviewer eArp),
“The paper is clearly written, with straightforward explanations” (Reviewer H57W).

2 **Method is practically valuable**, e.g.,
“conceptually simple and should be easy to implement in practice” (Reviewer QgqW),
“shows some nice computational scaling while yielding comparable accuracy” (Reviewer QgqW),
“offers a simple and efficient way to reduce supervision cost… achieving stable improvements even in resource-constrained settings” (Reviewer H57W).

3 **Experiment is comprehensive**, e.g.,
“comprehensively tested” (Reviewer xjoM),
“provide sufficient experimental details” (Reviewer eArp),
“empirical evaluation is extensive… Results are consistent across benchmarks” (Reviewer H57W).

4 **Paper is well written**, e.g.,
“The paper is well written and easy to follow” (Reviewer xjoM),
“The paper is clearly written” (Reviewer H57W),
“reproducibility should not be a problem” (Reviewer eArp).

5 We are especially **grateful to Reviewer eArp** for his/her sharp observations, as we focus on a solution and underlying principle for a specific scenarios, rather than a general discussion, which “**deepens the discussion depth of this paper**”.

---

> ### Author Response · Authors · 2025-11-21
> **Response to All Reviewers (2)**
>
> Besides, reviewers also provide several constructive comments and questions. Below, we first **respond to the common issues** raised across multiple points.
>
>
> **1 Comparison with ground-truth rewards and other unsupervised rewards methods.**
>
> We add experiments using both GRPO with ground-truth labels (noted as GRPO) and GRPO with majority-vote pseudo-labels (noted as mv-GRPO) for post-training. As shown in Tab.1, the first-repeat method performs comparably to majority vote, confirming that our lightweight selection strategy is as effective as standard self-consistency–based unsupervised baselines. The GRPO with ground truth rewards performs certainly better, but the performance gap narrows as model size increases. We hypothesize that larger models generate more reliable pseudo-answers, making first-repeat increasingly competitive to ground-truth rewards.
>
> In addition, we apply the First Repeat method to other preference optimization algorithms, including DrGRPO[1] and GPG[2]. DrGRPO is designed to achieve an unbiased optimization objective, while GPG directly optimizes the original reinforcement learning objective. Experimental results in Tab.5 show that our method can be effectively integrated into other preference optimization algorithm, consistently leading to performance improvements.
> Together, these supplementary experiments provide evidence for the robustness and effectiveness of our method.
>
> [1] Understanding R1-Zero-Like Training: A Critical Perspective
>
> [2] GPG: A Simple and Strong Reinforcement Learning Baseline for Model Reasoning
>
> **2 Design and analysis of hyperparameter tau**
>
> As discussed in Sec. 3.3 and Fig. 3, we observe a unique positive correlation with **phase-transition-like behavior **between average accuracy rewards and pseudo-answer accuracy. Once the average accuracy reward surpasses 0.8, pseudo-answer accuracy notably increases and becomes stable around 100%. Therefore, we set τ = 0.8 to control KL regularization against pseudo-label noise.
>
> In fact, our **experiments in Fig. 5 implicitly include a sensitivity analysis on tau**. The uns-GRPO w/o KL can be regarded as case with tau=0, and uns-GRPO w. KL can be regarded as case with tau=1. Those experiments provide a preliminary ablation study ton tau.
>
> For further rigor, we supplement our study with **additional experiments in Appendix D**, conducting a finer-grained sampling of τ as 0.2 and 0.5. The extended experiments reveal a clear pattern: as τ increases, model accuracy gradually improves, since uns-GRPO progressively applies KL regularization to low-confidence answers. Performance peaks at τ = 0.8 with an accuracy of 85.2%, then when τ reaches 1, accuracy drops to 84.2%, indicating that overly relying on the KL constraint restricts the model’s learning freedom from high-confidence answers.
>
> Therefore, both the confidence analysis and the ablation study consistently support τ = 0.8 as an effective setting. We state that, **our main contribution is not the specific form of τ**, **but the discovery of a phase-transition phenomenon** to confirm the label quality and **the demonstration that adaptive KL regulation** is beneficial for noisy label learning. We believe this discovery could provide more meaningful guidance for future research.
>
> **3 Conceptual novelty of first repeat method**
>
> Actually, in a broad sense, pseudo-labeling methods can be almost viewed as different instantiations of the self-consistency principle, including majority vote, cross labeling, and our first-repeat criterion. However, **our contribution goes beyond a minor engineering variation**.
>
> First, the pseudo-answer **generation process is fundamentally different**. First-repeat is an early-exit strategy, whereas majority vote is an exhaustive strategy that requires generating all samples before making a decision. This early-exit nature makes our approach **applicable to a wider range of reinforcement learning settings** (e.g., Monte-Carlo Tree Search), where stopping early after a reliable repetition can significantly reduce cost.
>
> Second, first-repeat **naturally extends to continuous-output task**s. For tasks such as object detection or keypoint localization, the output space is continuous, and majority voting becomes ill-defined because exact agreement almost never occurs. In contrast, our framework can be naturally extended by treating outputs within a small numerical tolerance as “repeats”, allowing first-repeat method applicable and efficient in continuous domains.
>
> Third, although first-repeat is a **folk wisdom** in everyday reasoning, to the best of our knowledge **we make the first attempt to formally analyze and theoretically justify** its rationality in unsupervised learning.
>
> We believe these aspects constitute a **conceptual contribution on unsupervised reinforcement learning**, rather than a purely engineering-oriented modification.

---

> ### Author Response · Authors · 2025-11-21
> **Response to All Reviewers (3)**
>
> In response to the reviewers’ comments, we have substantially **strengthened the revised paper by adding further studies**, including (1) comparisons with other reward methods in Tab.1, (2) analyses of first-repeat on larger models in Fig.7, (3) ablations studies on the τ parameter in Tab.4, (4) experiments integrating other GRPO variants in Tab.5, (5) statistics on wrong pseudo answer ratios of weak models in App.F, and (6) evaluations on code-generation and QA tasks.
>
> In addition, **we will provide detailed responses to each reviewer** to address the concerns and further improve the quality of this work.

---

### Author Response · Authors · 2025-11-27
**Awaiting for your further discussion**

Dear reviewers and chairs,

Thank you for your thoughtful reviews and constructive feedback. We have submitted our responses to all reviewer comments last week, addressing the identified concerns and providing additional analyses and clarifications.

At this stage, we respectfully invite the reviewers to revisit our rebuttal and engage in further discussion. We sincerely appreciate your time and efforts, and we **look forward to your further discussion and feedback**. Please let us know if any additional experiment or clarification would be helpful.

---

### Author Response · Authors · 2025-12-01
**Summary of Discussion**

Dear New Area Chair,

Thank you for your patience and hard work in this urgent situation.

Considering the urgent policy changes in the ICLR review procedure, and to assist your evaluation, we present a concise summary of the discussion.
We appreciate your time and look forward to your constructive comments.

## Summary of this work

This paper aims to enhance the reasoning abilities of large models using **unsupervised method** and assumes
the **first repeat answer** in a series of responses as the pseudo answer. Authors observe a **phase-transition-like consistency** between pseudo answer confidence and the average accuracy reward, and  **update the original KL regularization**
with an adaptive coefficient to protect the unsupervised learning from label noise.

## Summary of first round feedback

After detailed reviews from four reviewers, the paper has received a series of constructive comments with scores as 4,6,6,4. We are very grateful for the **recognition by reviewers**, including, **Motivation** is clear, **Method** is practically valuable, **Experiment** is comprehensive and **Paper** is well written.

We are **especially grateful to Reviewer eArp** for his/her sharp observations, as we focus on a solution and underlying principle for a specific scenarios, rather than a general discussion, which ***'deepens the discussion depth of this paper'***.

For detailed recognition from reviewers, please refer to **Response to All Reviewers (1)**.

## Common Issues

Reviewers also provide several constructive comments and questions, including **comparison with supervised  and other unsupervised methods**, **design and analysis of adaptive KL regularization** and **conceptual novelty** of first repeat method.

To address these concerns, we have supplemented the manuscript with additional experiments to validate the effectiveness and rationale of our method.
Moreover, we would like to stress that our **first repeat method is not a modification** of the majority vote strategy but is intrinsically distinct in its underlying mechanism. We **make the first attempt to formally analyze and theoretically justify its rationality in unsupervised learning**.

For detailed disscussion, please refer to **Response to All Reviewers (2)** and the related disscuss in **responses to each reviewer**.

## Awaiting for the discussion

Following the ICLR recommendations, **we submitted our responses to all comments before November 21** and have been waiting for subsequent discussion from the reviewers. At this stage, we would also **greatly appreciate your constructive feedback**, if the system permits.

Thank you again for your patience and time!

---

### Author Response · Authors · 2025-12-04
**Closing Statement (1)**

Dear Chair,

We sincerely thank the reviewers for their thoughtful and constructive feedback, and deeply understand and comply with ICLR's rule adjustments in this emergency situations. In response to the raised concerns, we have made substantial revisions that significantly strengthen the technical soundness, experimental rigor, and clarity of our work on unsupervised preference optimization with the first-repeat criterion.

For the convenience of the new AC and Chairs, we hereby state the important conclusions as followed.

---

### Author Response · Authors · 2025-12-04
**Closing Statement (2), Comparison with other post-training method**

We add experiments using other post-training methods, including GRPO with ground-truth labels (noted as GRPO) and GRPO with majority-vote pseudo-labels (noted as mv-GRPO) for post-training. As shown in **Tab.1 in Page 7** of our revised paper, the first-repeat method performs comparably to majority vote, confirming that our lightweight selection strategy is as effective as standard self-consistency–based unsupervised baselines. The GRPO with ground truth rewards performs certainly better, but the performance gap narrows as model size increases. We hypothesize that larger models generate more reliable pseudo-answers, making first-repeat increasingly competitive to ground-truth rewards.




| Model                         | Method   | AIME24 | MATH-500 | AMC23 | Minerva | OlympiadBench | Average.      |
|------------------------------|----------|--------|----------|-------|---------|----------------|---------------|
| Qwen2.5-0.5B-Instruct        | Baseline | 0      | 30.0     | 5     | 6.99    | 7.56           | 9.1           |
| Qwen2.5-0.5B-Instruct        | +GRPO    | 3.33   | 33.8     | 12.5  | 8.82    | 7.77           | 13.24         |
| Qwen2.5-0.5B-Instruct        | +mv-GRPO | 3.33   | 31.4     | 12.5  | 8.46    | 7.41           | 12.62         |
| Qwen2.5-0.5B-Instruct        | **+uns-GRPO**| 3.33   | 32.4     | 12.5  | 8.09    | 7.26           | 12.71 (↑ 3.61) |
| DeepSeek-R1-Distill-Qwen-1.5B| Baseline | 23.33  | 82.8     | 75.0  | 28.68   | 54.81          | 52.92         |
| DeepSeek-R1-Distill-Qwen-1.5B| +GRPO    | 33.33  | 86.6     | 75.0  | 32.35   | 52.44          | 56.54         |
| DeepSeek-R1-Distill-Qwen-1.5B| +mv-GRPO | 33.33  | 85.8     | 77.5  | 30.15   | 52.59          | 55.87         |
| DeepSeek-R1-Distill-Qwen-1.5B| **+uns-GRPO**| 30.0   | 85.2     | 80.0  | 31.62   | 52.74          | 55.91 (↑ 2.99)|
| Qwen2.5-7B-Instruct          | Baseline | 13.3   | 75.4     | 52.5  | 33.82   | 37.33          | 42.47         |
| Qwen2.5-7B-Instruct          | +GRPO    | 10.0   | 78.0     | 57.5  | 38.6    | 39.85          | 44.79         |
| Qwen2.5-7B-Instruct          | +mv-GRPO | 13.3   | 77.6     | 55.0  | 37.13   | 36.89          | 43.99         |
| Qwen2.5-7B-Instruct          | **+uns-GRPO**| 10.0   | 77.8     | 55.0  | 37.5    | 37.33          | 43.57 (↑ 1.10)|
| DeepSeek-R1-Distill-Qwen-7B  | Baseline | 56.67  | 94.0     | 87.5  | 39.34   | 67.11          | 68.92         |
| DeepSeek-R1-Distill-Qwen-7B  | +GRPO    | 56.67  | 94.8     | 95.0  | 41.54   | 67.26          | 71.05         |
| DeepSeek-R1-Distill-Qwen-7B  | +mv-GRPO | 53.33  | 94.8     | 92.5  | 41.48   | 68.3           | 70.08         |
| DeepSeek-R1-Distill-Qwen-7B  | **+uns-GRPO**| 56.67  | 94.6     | 95.0  | 40.44   | 67.85          | 70.91 (↑ 1.99)|

Overall, the experimental evidence confirms the reliability of our uns-GRPO method relative to alternative post-training approaches.

---

### Author Response · Authors · 2025-12-04
**Closing Statement (3), evaluation with other PO algorithms**

We further apply the First Repeat method to other preference optimization algorithms, including DrGRPO[1] and GPG[2]. DrGRPO is designed to achieve an unbiased optimization objective, while GPG directly optimizes the original reinforcement learning objective. Experimental results in **Tab.5 in Page 15** show that our method can be effectively integrated into other preference optimization algorithm, consistently leading to performance improvements.

| Model                          | Method       | AIME24 | MATH-500 | AMC23 | Minerva | OlympiadBench | Average.        |
|--------------------------------|--------------|--------|----------|-------|---------|----------------|-----------------|
| DeepSeek-R1-Distill-Qwen-1.5B | Baseline     | 23.33  | 82.8     | 75.0  | 28.68   | 54.81          | 52.92           |
| DeepSeek-R1-Distill-Qwen-1.5B | +GRPO        | 33.33  | 86.6     | 75.0  | 32.35   | 52.44          | 56.54 (↑ 3.62)  |
| DeepSeek-R1-Distill-Qwen-1.5B | +uns-GRPO    | 30.0   | 85.2     | 80.0  | 31.62   | 52.74          | 55.91 (↑ 2.99)  |
| DeepSeek-R1-Distill-Qwen-1.5B | +GPG         | 36.67  | 85.8     | 75.0  | 29.78   | 51.58          | 55.77 (↑ 2.85)  |
| DeepSeek-R1-Distill-Qwen-1.5B | +uns-GPG     | 30.0   | 85.0     | 77.5  | 29.41   | 52.44          | 54.87 (↑ 1.95)  |
| DeepSeek-R1-Distill-Qwen-1.5B | +DrGRPO      | 30.0   | 85.2     | 80.0  | 31.99   | 52.3           | 55.90 (↑ 2.98)  |
| DeepSeek-R1-Distill-Qwen-1.5B | +uns-DrGRPO  | 26.67  | 84.2     | 80.0  | 31.25   | 52.15          | 54.86 (↑ 1.94)  |


The supplementary experiments provide evidence for the robustness and effectiveness of our method.

[1] Understanding R1-Zero-Like Training: A Critical Perspective

[2] GPG: A Simple and Strong Reinforcement Learning Baseline for Model Reasoning

---

### Author Response · Authors · 2025-12-04
**Closing Statement (4), Study and Clarity of hyperparameter tau**

To mitigate the noise introduced by pseudo-labels, we design an adaptive KL regularization that dynamically adjusts regularization weight based on label confidence, and introduce a hyperparameter tau to distinguish whether a pseudo-answer is trustworthy.

We observe a unique positive correlation with **phase-transition-like behavior** between average accuracy rewards and pseudo-answer accuracy. Once the average accuracy reward surpasses 0.8, pseudo-answer accuracy notably increases and becomes stable around 100%. Therefore, we set τ = 0.8 to control KL regularization against pseudo-label noise.

For further rigor, we supplement our study with additional experiments in **Appendix D in Page 14**, conducting a finer-grained sampling of τ as 0.2 and 0.5. The extended experiments reveal a clear pattern: as τ increases, model accuracy gradually improves, since uns-GRPO progressively applies KL regularization to low-confidence answers. Performance peaks at τ = 0.8 with an accuracy of 85.2%, then when τ reaches 1, accuracy drops to 84.2%, indicating that overly relying on the KL constraint restricts the model’s learning freedom from high-confidence answers.

| τ        | 0    | 0.2  | 0.5  | 0.8  | 1    |
|----------|------|------|------|------|------|
| Accuracy | 81.8 | 82.2 | 84.0 | **85.2** | 84.2 |


Therefore, both the confidence analysis and the ablation study consistently support τ = 0.8 as an effective setting.

We state that, **our main contribution is not the specific form of tau, but the discovery** of a phase-transition phenomenon to confirm the label quality and the demonstration that adaptive KL regulation is beneficial for noisy label learning. We believe this discovery could provide more meaningful guidance for future research.

---

### Meta-Review · Area_Chair_QUAA · 2026-01-08

**Summary:**

This paper proposes an unsupervised RLVR framework, uns-GRPO. Firstly, the pseudo label is assigned based on the first repeated answer. Secondly, an adaptive KL regularization is used to mitigate the noise in the pseudo label. The authors demonstrated that this unsupervised approach achieves promising results compared to baselines.

The reviewers generally found the method to be intuitive and reasonable, but also raised significant concerns regarding the technical depth and the empirical evaluation of the paper. Specifically, the reviewers regarded the method as a heuristic extension of the majority voting pseudo-labeling method (H57W) and requested comparisons to more baselines (e.g., w/ ground-truth and using majority voting). In the author rebuttal, the authors provided more results but some major concerns remained unaddressed.

In addition, the submission didn't use the ICLR template (though this is not a decisive factor in the final recommendation).

**Reviewer Concerns:**

The concern regarding technical novelty and depth was not adequately addressed in the rebuttal. Also, the baseline results added by the authors during the rebuttal raised new questions that were not explained -- the authors showed that their first-repeat method performs comparably to majority vote, but didn't explain the reason (even not in a hand-wavy manner). Intuitively, the first-repeat method is a more efficient but simplified version of majority vote, and it in principle will underperform majority vote. This implied that the evaluation/training setup might have hidden the weakness of the proposed method (similar to the concerns raised by Reviewer QgqW), and the authors should test models beyond the Qwen family which has been reported to possess surprisingly good self-enhancement performance only based on self-confidence rather than ground-truth labels.

**Reviewer Scores:**

The reviewer scores would be unlikely to go up given that the core concerns were not well addressed in the author rebuttal and the new results presented in the rebuttal raised more concerns.

---

### Decision · Program_Chairs · 2026-01-26

Reject